# GMS-YOLO: An Algorithm for Multi-Scale Object Detection in Complex Environments in Confined Compartments

**DOI:** 10.3390/s24175789

**Published:** 2024-09-05

**Authors:** Qixiang Ding, Weichao Li, Chengcheng Xu, Mingyuan Zhang, Changchong Sheng, Min He, Nanliang Shan

**Affiliations:** 1National Key Laboratory of Electromagnetic Energy, Naval University of Engineering, Wuhan 430033, China; dqx312@alumni.nudt.edu.cn (Q.D.); liweichao23@nue.edu.cn (W.L.); zmyxinyang@126.com (M.Z.); shengcc_nue@163.com (C.S.); nanliang@stu.xmu.edu.cn (N.S.); 2Department of Electrical Engineering and Automation, Wuhan University, Wuhan 430072, China; whuhemin@whu.edu.cn

**Keywords:** confined compartments, complex environments, multiscale object detection, YOLOv8, volumetric light weighting

## Abstract

Many compartments are prone to pose safety hazards such as loose fasteners or object intrusion due to their confined space, making manual inspection challenging. To address the challenges of complex inspection environments, diverse target categories, and variable scales in confined compartments, this paper proposes a novel GMS-YOLO network, based on the improved YOLOv8 framework. In addition to the lightweight design, this network accurately detects targets by leveraging more precise high-level and low-level feature representations obtained from GhostHGNetv2, which enhances feature-extraction capabilities. To handle the issue of complex environments, the backbone employs GhostHGNetv2 to capture more accurate high-level and low-level feature representations, facilitating better distinction between background and targets. In addition, this network significantly reduces both network parameter size and computational complexity. To address the issue of varying target scales, the first layer of the feature fusion module introduces Multi-Scale Convolutional Attention (MSCA) to capture multi-scale contextual information and guide the feature fusion process. A new lightweight detection head, Shared Convolutional Detection Head (SCDH), is designed to enable the model to achieve higher accuracy while being lighter. To evaluate the performance of this algorithm, a dataset for object detection in this scenario was constructed. The experiment results indicate that compared to the original model, the parameter number of the improved model decreased by 37.8%, the GFLOPs decreased by 27.7%, and the average accuracy increased from 82.7% to 85.0%. This validates the accuracy and applicability of the proposed GMS-YOLO network.

## 1. Introduction

In confined compartments such as compartments where bolts, binds, and plugs, among other important fasteners, may loosen or be missing, there is a direct impact on the overall system. In addition, there is also the potential for localized pooled water and surface damage to various equipment in the compartment. Hence, it is necessary to employ accurate and efficient detection methods. Traditional manual inspection methods are costly and inefficient and are unable to effectively inspect confined compartments, which poses potential safety risks. In recent years, the application of computer vision and deep learning techniques for intelligent inspection has become a research hotspot. When utilizing artificial intelligence and deep learning methods for various inspection tasks, various object detection algorithms have become representative methods.

For image object detection, traditional methods are predominantly based on sliding windows and manual feature extraction, which often suffer from high computational complexity and robustness issues in complex scenarios [1]. In recent years, deep learning-based object detection algorithms have been widely recognized across various fields [2]. Starting from the R-CNN algorithm proposed in [3], numerous deep learning-based image object detection algorithms have emerged since 2014, including Fast R-CNN [4], Faster R-CNN [5], SPPNet [6], and YOLO [7], among others. Compared to traditional object detection algorithms, deep learning-based object detection algorithms exhibit advantages such as high speed, strong accuracy, and excellent robustness in complex conditions [8].

Object-detection algorithms have also seen rapid development in various tasks [9,10,11], including various inspection tasks. One of the typical scenarios that has received considerable research attention is the detection of insulators in transmission line scenarios. The uniqueness of this scenario lies in the complex environment in which the insulator is situated. Furthermore, the diversity in the sizes of insulator defects and failures can lead to a decline in model detection accuracy. YOLO (You Only Look Once) series algorithms, as representative single-stage target detection algorithms, perform exceptionally well in visual inspection scenarios and are widely applied. In reference [12], by capturing edge information using maximum pooling and detailed texture information using average pooling, the accuracy of the model in detecting insulators and their defects in complex backgrounds has been enhanced. In reference [13], a rotated candidate box was designed to improve the quality of prediction boxes and reduce irrelevant backgrounds. In reference [14], five detectors were used for object localization and detection, thereby employing more detection outputs for comprehensive object location. In reference [15], a CSPResNeSt feature-extraction backbone was designed in the CSPNet backbone network to address the problem of insulator detection in complex environments. Therefore, we opt to use YOLO as the primary framework of our study, optimizing the structure of each network phase to enhance the final performance of the model.

For visual object detection in confined compartments, its uniqueness lies in the complexity of the environment within the cabin and the diverse categories of inspection targets. Specifically, there is only limited artificial lighting within the cabin, which can lead to phenomena such as reflections, shadows, and uneven brightness in the images. The impact of this in a background with various equipment and cables is more severe. The inspection targets also encompass various categories such as fasteners (bolts, binds, plugs, etc.), various types of foreign objects, puddles, and surface damage on various equipment, with some small targets and targets that closely match the background color easily blending with shadows and backgrounds, making detection difficult. Additionally, the scale of the targets varies greatly, with their width-to-length ratios constantly changing and sizes varying, which can lead to significant interference in the detection of inspection targets. Lastly, in actual deployment in specific scenarios, the performance of edge-computing devices is relatively low, requiring the computational complexity of the model to be as low as possible.

In recent years, researchers have proposed numerous methods to address the problem of multi-scale object detection in complex environments. Based on deep learning, object detection methods depend on backbone networks to acquire high-dimensional features. However, in the inspection scenario, the size and proportions of the target vary greatly, necessitating the use of multi-scale features to more effectively represent features. Lin et al. [16] were inspired by the pyramid structure of handcrafted feature extraction and proposed the feature pyramid network (FPN), which can aggregate high-resolution low-level features with low-resolution high-level features. Subsequently, PANet [17], NAS-FPN [18], ASFF [19], and BiFPN [20] were proposed, achieving good results in object detection tasks. Cheng et al. [21] enhanced features before fusion using dual attention mechanisms, allowing the network to focus on the obvious characteristics of the object. Zhang and Shen’s feature enhancement module [22] is similar to Cheng’s, also employing attention mechanisms in both spatial and channel dimensions to enhance features. However, these methods still face issues such as insufficient feature extraction or insufficient capability in multi-scale feature fusion. Therefore, we designed a novel and superior GhostHGNetv2 network as the feature-extraction network of our model, aiming to enhance the features and utilize the Multi-Scale Attentive Module (MSCA) for the feature fusion part of the model to optimize its performance. This mitigated the shortcomings of insufficient feature extraction and insufficient capability in multi-scale feature fusion.

In the context of deployment applications for cabin inspection scenarios, lightweightness is a crucial metric to measure the detectors. This necessitates optimizing precision within limited computational resources and model volume. There are two common methods for making networks lightweight. The first is termed model compression, which involves pruning parameters that fall below the threshold set by the design filter algorithm [23,24,25,26]. Any model can be reduced in size by pruning. The second approach is to use lightweight convolutional networks to optimize the model structure. It involves designing more efficient computation methods for the network. MobileNet [27], ShuffleNet [28], and GhostNet [29] use deep convolutional (DWConv) and/or grouped convolutional to extract spatial features. However, these modules have not been widely adopted in this context, and thus, we refer to the practice of lightweight convolutional networks, introduce lightweight convolutional modules into our model, and design an efficient detection head SCDH to reduce the overall model size. This decreases the memory footprint and inference time of the target detection algorithm.

To address the aforementioned issues and challenges, we propose an improved strategy (GMS-YOLO) to solve the multi-object inspection problem in confined compartments. GMS-YOLO can accurately identify targets such as fasteners, water accumulation, foreign objects, and surface defects in complex scenarios while significantly reducing the computational complexity of the model. The main contributions and innovations of this paper are as follows:

(1) Addressing the issues and requirements of edge deployment in complex environments, we adopt a novel GhostHGNetv2 backbone network to replace the original DarkNet backbone network. This modification utilizes Ghost Convolution to improve HGNetv2 and results in GhostHGNetv2, reducing the computational load of redundant feature maps while incorporating channel features. GhostHGNetv2, while reducing the overall volume and computational complexity of the model, can extract more abundant feature information, thus better distinguishing between the object and background and improving the final detection accuracy.

(2) Addressing the issue of target scale variation and the need for edge deployment, we first introduce the lightweight Multi-Scale Convolutional Attention (MSCA) module as the high-level semantic feature refinement module to capture multi-scale contextual information and guide the feature fusion process. Secondly, we design a novel lightweight detection head, Shared Convolutional Detection Head (SCDH). It employs shared convolution to aggregate feature maps of different sizes, reducing redundant parameters while achieving the effect of multi-scale feature fusion.

(3) Addressing the multi-object inspection issue in confined compartments, we constructed a mixed dataset composed of data collected from field shooting and publicly available data to support the progress of this study. The algorithms studied in this paper will be experimentally validated and analyzed in this dataset.

## 2. Method

### 2.1. Overview of the Network Framework

The overall structure of the GMS-YOLO algorithm is shown in Figure 1. In this figure, the improved HGNetv2 backbone network, which is composed of Ghost Convolution-enhanced Ghost HGBlocks, is used for feature extraction to address object detection problems in complex backgrounds. HGNetv2 is an efficient feature-extraction network that can extract more abundant feature information, thereby enhancing object detection accuracy. Furthermore, after Ghost Convolution’s improvements, the network becomes more lightweight and preserves accuracy. In the fusion area, we modified the original PAFPN structure to add a semantic feature refinement structure before high-level semantic fusion. We utilized the MSCA module to refine high-level semantic features, enabling the network to capture the target from a global perspective more adeptly. At the same time, the features from the low-level network are guided by the fusion after the network, thereby improving the overall feature fusion performance. Lastly, in the detection head, we designed a lightweight detection head based on shared convolution, finally achieving a higher accuracy at a smaller parameter size and GFLOPs.

### 2.2. Feature Extraction Network Based on an Enhanced HGNetv2

In this study, due to the presence of complex backgrounds in many scenarios, detecting small objects or objects that share a similar style with the adjacent background presents certain challenges. This section primarily focuses on this issue.

In object detection algorithms, a well-trained backbone feature-extraction network can better distinguish between the differences between the background and the target to be detected, which is often used to address challenges in detecting scenes with complex backgrounds. The extracted features are more conducive to subsequent models learning effective targets from the features. Following the success of PP-HGNet in reference [30], this study initially proposes a model improvement research based on the iteration version of HGNetv2, which has achieved excellent performance in PP-HGNet. HGNetv2 consists of four stages, each of which is constructed with a HGBlock. The input first enters the first layer of Stage 1, which is the HGStem module, effectively reducing the number of parameters and computational load. The four stages all use learnable downsampling layers (LDS Layer). Stage 1 does not include learnable downsampling layers, whereas stages 2 to 4 do. Traditional downsampling methods, such as max pooling and average pooling, have fixed parameters and cannot be changed [31,32]. The uniqueness of the learnable downsampling layer lies in its ability to dynamically perform downsampling based on input features, ultimately enhancing network performance and accuracy. The detailed network structure is shown in Figure 2.

The backbone network used in this study is GhostHGNetv2, an improvement of HGNetv2. The overall structure of the network is shown in Figure 3. The HGStem module remains unchanged, with the learnable downsampling layer (LDS Layer) using Depthwise (DW) Convolution. The Ghost Convolution module replaces the convolution module in the original HGBlock, transforming it into a GhostHGBlock module. Since this is a task of object detection, the network directly connects to the SPPF module in the YOLOv8 network for subsequent feature fusion preparation.

In our GhostHGNetv2 network, the structure of the HGStem module is shown in Figure 3. The process is represented by the following formulas:(1)$$Y_{1}={\mathrm{MaxPool}}_{2\times 2}^{1}\left[F_{3\times 3}^{2}\left(X\right)\right]$$
(2)$$Y_{2}=F_{2\times 2}^{1}\left\{F_{2\times 2}^{1}\left[F_{3\times 3}^{2}\left(X\right)\right]\right\}$$
(3)$$Y=F_{1\times 1}^{1}\left\{F_{3\times 3}^{2}\left[\mathrm{Concat}\left(Y_{1},Y_{2}\right)\right]\right\}$$
where *X* represents the input three-channel image data and *Y* represents the feature map output by the module. $F_{3\times 3}^{2}$, $F_{2\times 2}^{1}$, and $F_{1\times 1}^{1}$ indicate the general convolution operation with convolution kernel sizes of 3, 2, and 1, respectively, with convolution strides of 2, 1, and 1, respectively. ${\mathrm{MaxPool}}_{2\times 2}^{1}$ represents the 2 × 2 window size, stride 1 maximum pooling operation.

Additionally, another network GhostNet has achieved leading effects in terms of feature extraction. The focus is mainly on leveraging both primary and secondary channels. The primary channel generally captures the main information and features of an image, whereas the secondary channel, with fewer convolutional layers, targets the details and low-level features. The secondary channel acts as a regularization mechanism, aiding in the prevention of overfitting. Additionally, the secondary channel employs deep separable convolution, which reduces the model’s weight, parameters, and computational demands. Ultimately, the output from the secondary channel is combined with that of the primary channel to produce the final feature representation. The schematic diagram of the Ghost Convolution process within the GhostNet is shown in Figure 4.

The Ghost Convolution process can be represented as follows: (4)$$Y_{1}=F_{1\times 1}^{1}\left(X\right)$$
(5)$$Y=\mathrm{Concat}\left[Y_{1},D_{5\times 5}^{1}\left(Y_{1}\right)\right]$$
where $D_{5\times 5}^{1}$
represents a depth convolution operation with a kernel size of 5 × 5, stride 1, and group number equal to the number of channels.

Inspired by GhostNet and Ghost Convolution, we propose the GhostHGBlock, incorporating the Ghost Convolution module into the HGBlock in the HGNetv2 backbone. We replace the Light Convolution module in the HGBlock with the Ghost Convolution module, achieving better feature extraction while also making the model lighter, with fewer parameters and computational requirements. The schematic diagram of the GhostHGBlock module and its process is shown in Figure 5. The formula for the process is as follows: (6)$$\begin{array}{cc} Y_{1} & =G\left(X\right),\\ Y_{2} & =G\left(Y_{1}\right),\\ \; & \ldots \\ Y_{n} & =G\left(Y_{n-1}\right) \end{array}$$
(7)$$Y=F_{1\times 1}^{1}\left\{F_{1\times 1}^{1}\left[\mathrm{Concat}\left(Y_{1},Y_{2},\ldots ,Y_{n}\right)\right]\right\}$$
where G represents the Ghost Convolution operation mentioned above.

### 2.3. Multi-Scale Convolutional Attention

In this study, there are certain targets that are fixed in shape, yet their detection is disrupted due to inconsistent directions in images. Additionally, there are targets that have variable shapes, which can also pose a challenge to the detection task.

Addressing the critical issue of the variable aspect ratio of the target to be detected, we introduced the efficient Multi-Scale Convolutional Attention (MSCA) [33] as the high-level semantic feature refinement module. It enables the network to capture the target from a holistic perspective with greater proficiency. Moreover, after fusion, it provides guidance for the feature maps of the low-level network, enhancing the overall robustness of the network in detecting targets with variable aspect ratios.

The attention mechanism is an adaptive selection process aimed at focusing the network on important parts. Generally, it can be divided into two types [34], including channel attention and spatial attention. These different types of attention play different roles. For instance, spatial attention primarily focuses on important spatial regions [35,36,37,38]. Contrarily, using channel attention allows the network to selectively focus on important objects, which has been proven to be crucial in previous research [39,40,41]. The MSCA module utilizes the Large Kernel Attention (LKA) mechanism to establish channel and spatial attention while considering the role of multi-scale feature aggregation to achieve better results.

MSCA consists of three parts, as shown in Figure 6: a deep convolution for local information aggregation, a multi-branch deep Strip Convolution for capturing multi-scale context, and a 1 × 1 convolution to simulate the relationships among different channels. The 1 × 1 convolution’s output is directly used as the attention weight, to rebalance the input of MSCA. The aforementioned process is formulated as follows: (8)$$Branch_{0}=D_{5\times 5}^{1}\left(X\right)$$
(9)$$Branch_{n}=D_{m\times 1}^{1}\left[D_{1\times m}^{1}\left(Branch_{0}\right)\right],n=1,2,\ldots$$
(10)$$Att=F_{1\times 1}^{1}\left(\sum_{i=0}^{3}Branch_{i}\right)$$
(11)Y=Att⊗X
where *X* represents the input feature, Att and *Y* represent the attention map and output. $Branch_{i},i\in 0,1,2,3$ represents the branch in Figure 6, and *Branch*_0_ represents the identity connection. According to reference [42], in each branch, we approximate the standard deep convolution with 2 convolutional layers using large kernels. Here, the kernel sizes of each branch are set to 7, 11, and 21, respectively. We choose to use the Strip Convolution for two reasons. Firstly, the Strip Convolution is lightweight. To approximate a 7 × 7 standard 2D convolution, we only need a pair of 7 × 1 and 1 × 7 convolutions. Secondly, in specific detection scenarios, there exist some objects that are drawn as stripes, such as clamps, coils, and screwdrivers. Therefore, Strip Convolution can serve as an adjunct to Grid Convolution, aiding in the extraction of strip features [42,43].

### 2.4. Shared Convolutional Detection Head

In this study, many targets themselves vary in size considerably, and the shooting distance may not be consistent, leading to a significant difference in target size within the same category in images. This can also interfere with object detection tasks.

In addition, the YOLO series often uses separate object detection heads on different feature layers, resulting in a low utilization of model parameters. This is because the features of objects detected at different feature layers should be comparable in relative scale size, and academic detection models such as RetinaNet [44] and FCOS [45] support this with shared parameters.

To address the issues of inconsistent target size, model network redundancy, and low parameter utilization in YOLOv8, we redesigned the detection head part of the model. We proposed a Shared Convolutional Detection Head (SCDH).

Firstly, the Shared Convolutional Detection Head (SCDH) aggregates the feature maps of different sizes by using shared convolution, thereby obtaining a more accurate detection result by integrating the features of three sizes. At the same time, it can significantly reduce the number of parameters, making the model more lightweight, particularly on devices with resource constraints. Furthermore, since the statistical differences between the feature maps at different levels still exist, normalization layers are still necessary. Although BN can be introduced directly into the detection head-sharing parameters, its mean value will be erroneous due to the sliding average. Although GN improves the performance of detection head localization and classification in the reference [45], its introduction will increase inference overhead [46]. Therefore, we refer to NAS-FPN [18] and have the detection head share convolutional layers with BN, which is separately computed. Finally, to address the issue of inconsistent scales of objects detected by each detection head, we use scale layers to scale the features. The aforementioned structure is shown in Figure 7, where the blue region represents the parameter-sharing portion.

## 3. Experiment

### 3.1. Dataset Construction and Experimental Environment

To achieve the recognition of multiple objects in a confined compartment, this study established a corresponding dataset. This dataset is a mixed dataset with 5616 images, composed of image data collected from actual scenes and various publicly available image data (including public datasets such as COCO128). The dataset consists of eight categories: bolts, two types of bind, two types of plug, water, foreign objects, equipment surface damage, corresponding to the label terms of “bolt”, “a-bind”, “n-bind”, “a-plug”, “e-plug”, “water”, “fod”, and “flaw”. Notably, for categories such as water, foreign objects, and equipment surface damage that are not conducive to extensive deployment in the field, this dataset was obtained by collecting and integrating image data from scenes and public datasets and using instance-augmented image-augmentation methods. The final image data include 18,669 instances. Specifically, we initially collected 1351 images on-site, then expanded the dataset to 4630 images by incorporating data from the public COCO128 dataset and web sources. Finally, we conducted separate data augmentation for categories with fewer instances in the on-site collected images (including, but not limited to, random pixel discarding, sharpening, brightness adjustment, random hue changes, image flipping, and rotation) to obtain the final dataset of 5616 images. The detailed dataset data are shown in Table 1. Some sample content is shown in Figure 8.

The experiment was executed on a Windows 10 Professional system, utilizing Python 3.10 and CUDA 11.5. The deep learning model was trained and tested using PyTorch 1.11.0. A single NVIDIA GeForce RTX 3090 GPU with 24 GB of memory was employed for the training process. Detailed training configurations are presented in Table 2. The settings for the training include 300 epochs, a batch size of 16, an initial learning rate of 0.01, and the use of Stochastic Gradient Descent (SGD) as the optimizer.

### 3.2. Evaluation Indicators

Common performance metrics used in object detection tasks include parameters, computational complexity, recall rate, precision, F1-score, average precision (AP), and mean average precision (mAP). Parameters and computational complexity are metrics to describe the overall size and computational burden of the model. Computational complexity corresponds to time complexity, and parameters correspond to space complexity, which is based on the length of network execution time and the amount of memory occupied by the GPU. The recall rate measures the percentage of true positive samples within all identified positive samples. Precision, on the other hand, measures the percentage of identified positive samples within all true positive samples. These two metrics are often interrelated. A high recall rate may result in lower precision and more false positives. Conversely, high precision may lead to a lower recall rate and more missed detections. The F1-score, which is the harmonic mean of precision and recall, provides a comprehensive evaluation of the model’s performance by taking both metrics into account. The formulas for calculating recall rate, precision, F1-score, AP, and mAP are as follows: (12)$$Recall=\frac{T_{P}}{T_{P}+F_{N}}$$
(13)$$Precision=\frac{T_{P}}{T_{P}+F_{P}}$$
(14)$$F_{1}\text{-}score=\frac{2\times Precision\times Recall}{Precision+Recall}$$
(15)$$AP=\int_{0}^{1}P\left(r\right)dr$$
(16)$$mAP=\frac{1}{n+1}\sum_{i=1}^{n}AP_{i}$$
where $T_{P}$ stands for True Positive, $F_{P}$ for False Positive, and $F_{N}$ for False Negative. The P-R curve represents the function image encompassed by the recall and precision rates, while n denotes the number of defect categories. By combining the recall and precision rates for each detection, we obtain the P-R curve. AP is then calculated through interpolation, and mAP is ultimately computed.

### 3.3. Experimental Analysis

#### 3.3.1. Contrast Experiment

In the study of multi-object recognition in confined compartments, we initially designed an experiment to compare the performance of existing basic object detection algorithms. Considering the subsequent deployment at the edge and the need for accuracy, we primarily used YOLO series models with lighter weights and another representative RT-DETR series model for comparison experiments.

The experimental results are shown in Table 3. In terms of precision, most YOLO series models outperform RT-DETR series models, while the YOLO series models generally demonstrate a significant advantage in terms of model size and computational load.

Within the YOLO series, YOLOv5s and YOLOv8s both performed exceptionally well. YOLOv5s has a smaller model size and lower computational load than YOLOv8s, while YOLOv8s achieves higher Precision, Recall, and mAP compared to YOLOv5s. Considering the higher demand for accuracy and the relatively low model size and computational load of YOLOv8s, the subsequent experiments in this study mainly focus on improving and innovating on the YOLOv8s base model.

Subsequently, following the algorithm improvements, we also compared a few other improved algorithms. The experimental results are shown in Table 4. It can be observed that, within this scenario, for those algorithms with similar model size and computational complexity to the algorithm proposed in this paper (MFI-YOLO [47], BL-YOLOv8 [48]), our method exhibits higher accuracy and a lighter model size. For those algorithms with lighter model sizes (Insu-YOLO [49]), our method demonstrates a significant advantage in terms of accuracy. For those algorithms with relatively high accuracy (CS-YOLO [50]), our method maintains accuracy superiority while achieving a model size one to two times lighter. While comparing these improved algorithms, we also applied the proposed method in this paper to the YOLOv5s model, which is comparable in performance to YOLOv8s. This application demonstrated significant performance enhancements and a reduction in model size, fully validating the superiority and applicability of the algorithm proposed in this paper.

#### 3.3.2. Ablation Experiment

This paper conducts ablation experiments based on the YOLOv8s model, comparing the merits of each step. The main aspects can be divided into two categories: comparative analysis based on specific categories and comparative analysis based on the overall system.

(1) Comparative analysis based on specific categories: after the model was improved, it showed significant improvements for a majority of categories, as shown in Table 5. We will analyze the specific effects of each improved module in detail below.

Firstly, in response to the issue of complex backgrounds, the GhostHGNetv2 module plays a role in differentiating complex backgrounds from the objects to be detected, making it easier to detect the previously difficult-to-detect targets due to background interference. Specifically, bolt, a-bind, a-plug, and flaw, these four types are either small in size or easily blend into the background, making them susceptible to interference from complex backgrounds. After the improvement, the accuracy rates of these four categories increased by 0.4%, 2.2%, 3.5%, and 0.9%, respectively, validating the effectiveness of the module.

Secondly, in addressing the issue of variable aspect ratios in the target, the MSCA module plays a role in refining the semantic features at the higher level of the network, making it more adept at capturing the target from a global perspective. Simultaneously, it provides guidance to the lower-level network’s feature maps, thereby enhancing the overall robustness of the network in detecting targets with variable aspect ratios. Specifically, a-bind, a-plug, e-plug, and water, which either exhibit variable aspect ratios due to arbitrary plug orientation or due to diverse target shapes, are displayed in a variable aspect ratio in the image data. After the improvements, the accuracy rates of these four categories increased by 0.5%, 0.7%, 0.6%, and 0.8%, respectively, fully validating the efficacy of the module.

Lastly, in addressing the issue of variable sizes in the target, the SCDH module plays a role in aggregating the features of three-sized feature maps and ultimately outputting the detection result, thereby enhancing the robustness of the network in detecting targets of varying sizes. Specifically, bolt, a-bind, a-plug, water, and fod, which all arise due to targets with varying sizes, pose detection challenges. After the improvements, the accuracy rates of these five categories increased by 0.7%, 2.3%, 3.0%, 1.7%, and 0.7%, respectively, fully validating the efficacy of the module.

Additionally, we have observed that among all eight categories, the detection accuracy and recall rates for the targets classified as bolt, a-plug, and flaw are significantly inferior to those of the other categories. Upon analysis, it was found that this is primarily due to limitations in the dataset and the scenarios. Specifically, these categories are more prone to occlusion or blending with shadows due to lighting conditions. Furthermore, the flaw category includes a large number of damage instances collected from various scenes on the internet, which makes it difficult for the model to learn effective features.

(2) Comparative analysis based on the whole: the improved model performs better in all aspects, as shown in Table 6 below. We will now elaborate on the specific effects of each improved module.

Firstly, our GhostHGNetv2 is used as the backbone network of the model, resulting in a 1.1% increase in mAP while also reducing the model’s total parameter size by 2.8 m and FLOPs by 5.5 G. This improvement is primarily due to the better consideration of feature reuse reduction-related feature mapping in GhostHGNetv2 and the reduction in redundant computation operations.

Secondly, we integrate MSCA, the Multi-Scale Convolutional Attention, into the network, which results in a 0.6% increase in mAP with a minimal increase in parameter size and FLOPs. This improvement is primarily due to the use of a large kernel strip convolutional attention mechanism in MSCA to establish channel and spatial attention, along with the consideration of the role of multi-scale feature aggregation in achieving better detection results.

Thirdly, we integrate our designed SCDH module as the final detection head module in the model, resulting in a 1.3% increase in mAP while reducing the parameter size by 1.7 m and FLOPs by 2.6 G. This improvement primarily stems from the use of shared convolution in SCDH to integrate low-, medium-, and high-level feature maps, thereby enhancing the extraction of features in GhostHGNetv2 and strengthening the guidance of MSCA for feature fusion. Furthermore, it reduces redundant computational operations.

Finally, our designed GMS-YOLO model achieved a reduction in parameter size of 37.8%, a decrease in GFLOPs of 27.7%, and an increase in the average accuracy from 82.7% to 85.0%.

#### 3.3.3. Visualized Analysis

In Figure 9, we present a comparison between the improved model and the original model in terms of loss curves and metrics curves. It can be observed that while the improved model shows little difference from the original model in terms of box loss and class loss, it exhibits a significant advantage in distribution focal loss. Additionally, the improved model outperforms the original model in all performance metrics. Therefore, the effectiveness of the model improvement is validated from the perspectives of both loss curves and performance metric curves.

In Figure 10, we demonstrate the results of the detection performed by the improved and original models on a subset of the validation dataset. These four images all depict the situation of detecting multiple categories of targets under this scenario. It is evident that the original YOLOv8s model suffers from an issue of false detections (a-bind) in the first image. It also suffers from false detections (fod and bolt), missing detections (n-bind and a-plug), and the presence of redundant detection boxes in the second image. In the third image, it suffers from false detections (bolt) as well as missing detections (bolt and fod). In the fourth image, it suffers from false detections (fod). Compared to the original model, our improved model has made significant improvements in avoiding false detections and missing detections, thus facilitating more effective detection of the target. Moreover, from the perspective of heat maps, it can also be observed that our method exhibits superior performance in feature focusing, as it is more adept at drawing attention to the target of interest rather than the background or irrelevant objects. Therefore, the effectiveness of the model improvement has been validated from the perspective of validation dataset images.

## 4. Conclusions

This paper presents a novel algorithm based on YOLOv8, referred to as GMS-YOLO, for solving the inspection problem of multiple targets in confined compartments. Firstly, a high-performance improved backbone network, GhostHGNetv2, is employed as the feature-extraction network to enhance the target feature-extraction capability under complex backgrounds in confined compartments while reducing the model volume and computational complexity. Secondly, a lightweight multi-scale attention mechanism, MSCA, is introduced as the semantic feature refinement module to capture multi-scale contextual information and guide the feature fusion process. This improves the accuracy of multi-scale object detection by reducing the number of network model parameters. Finally, a novel lightweight detection head, SCDH, is designed. It employs shared convolution to aggregate different-sized feature maps, reducing redundant parameters and computational cost while achieving the effect of multi-scale feature fusion. A mixed dataset composed of image data collected from the field and public datasets is constructed, and experiments are conducted on this dataset. The experimental results demonstrate that GMS-YOLO outperforms current classic object detection models and improved models in various aspects, demonstrating its superior performance.

Overall, the proposed model exhibits superior performance in detecting various targets in confined compartments, while also reducing computational complexity. This makes it a feasible solution for practical deployment. In the cabin inspection dataset constructed in this paper, unknown foreign objects and abrupt lighting conditions could potentially lead to performance degradation. In the future, we will continue to collect more data on unknown foreign objects and images under different lighting conditions. Additionally, we will expand the data on categories with fewer instances, aiming to continually enrich and improve the dataset. Furthermore, we will also investigate novel high-performance feature extraction, fusion methods, and lightweight approaches in the future, with the aim of achieving better performance.

## Figures and Tables

**Figure 1 sensors-24-05789-f001:**
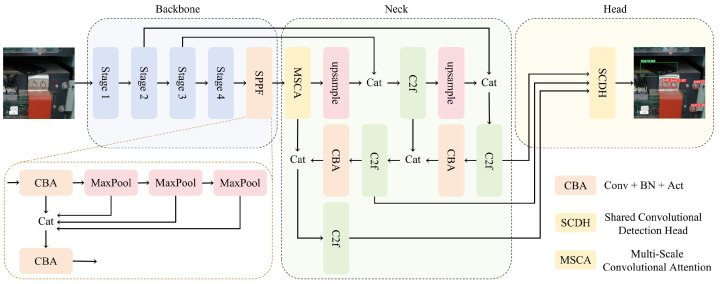
Overview of the GMS-YOLO network framework.

**Figure 2 sensors-24-05789-f002:**
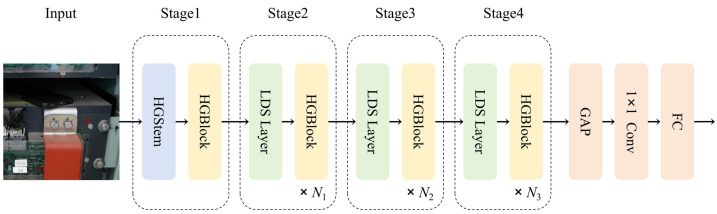
The network structure of HGNetv2.

**Figure 3 sensors-24-05789-f003:**
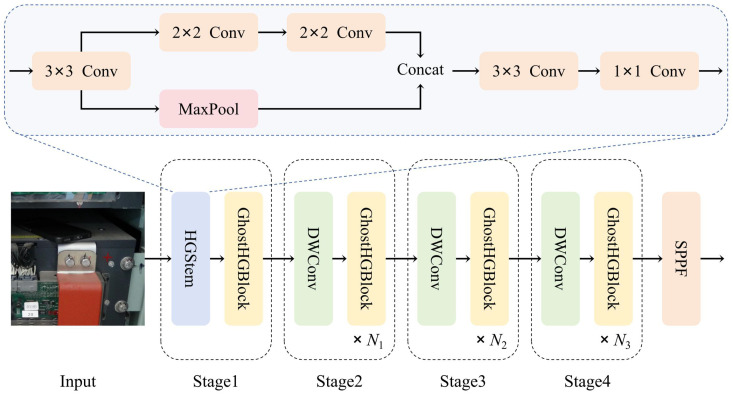
The network structure of GhostHGNetv2.

**Figure 4 sensors-24-05789-f004:**
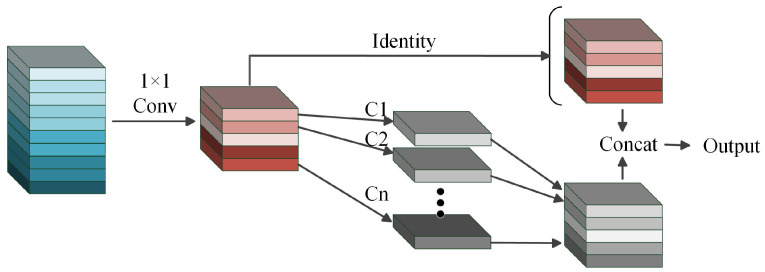
The structure of Ghost Convolution module.

**Figure 5 sensors-24-05789-f005:**
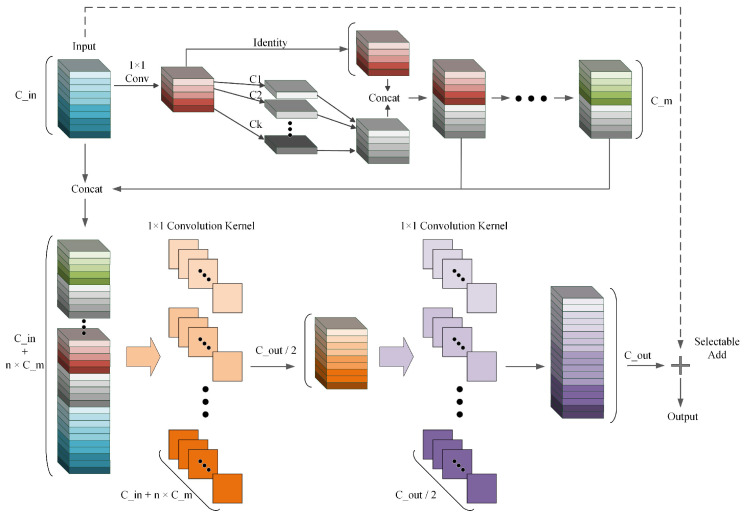
The structure of GhostHGBlock module.

**Figure 6 sensors-24-05789-f006:**
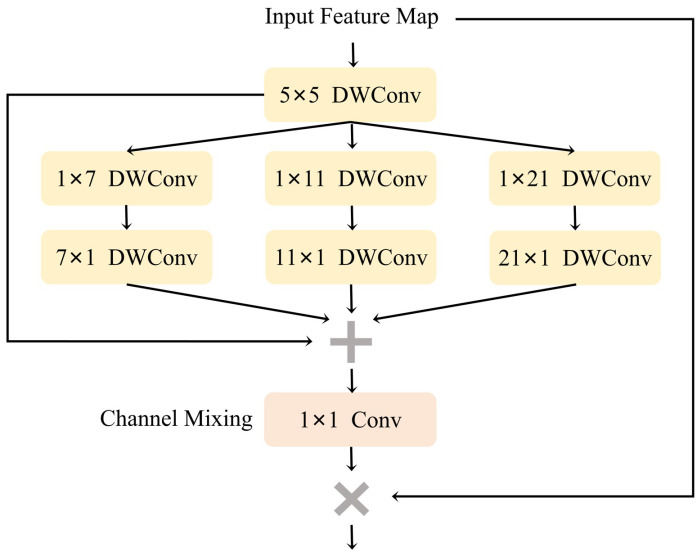
The structure of MSCA module.

**Figure 7 sensors-24-05789-f007:**
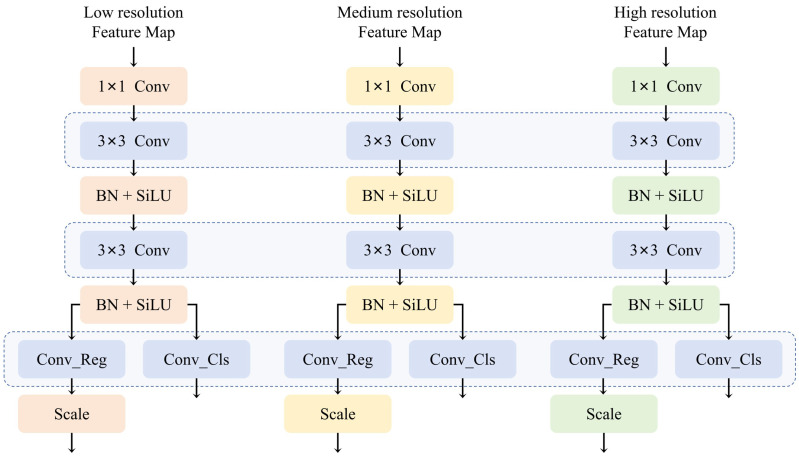
The structure of SCDH module.

**Figure 8 sensors-24-05789-f008:**
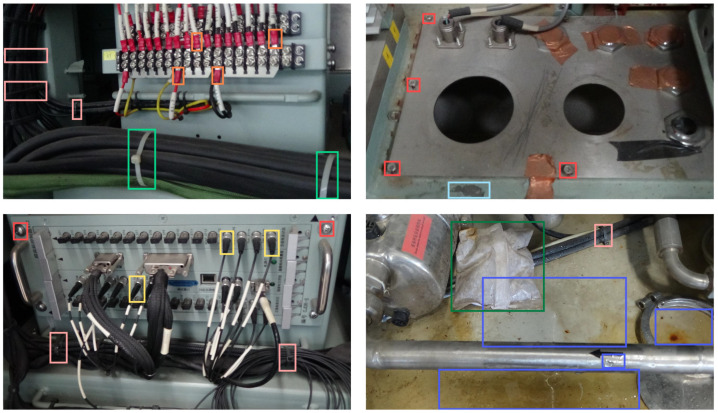
Sample presentation of the dataset. These figures showcase some typical scenes and targets within our dataset, specifically focusing on the equipment and environment in this confined cabin setting, which includes all categories that need to be detected. The red detection boxes represent bolt, the orange detection boxes represent e-plug, the yellow detection boxes represent a-plug, the dark green detection boxes represent fod, the light green detection boxes represent a-bind, the pink detection boxes represent n-bind, the light blue detection boxes represent flaw, and the dark blue detection boxes represent water.

**Figure 9 sensors-24-05789-f009:**
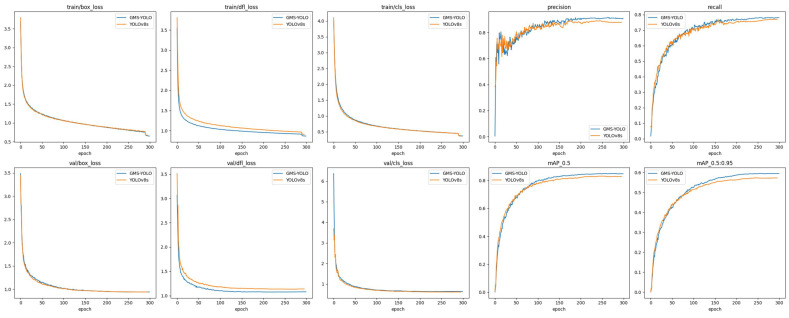
Comparison of loss curves and metric curves.

**Figure 10 sensors-24-05789-f010:**
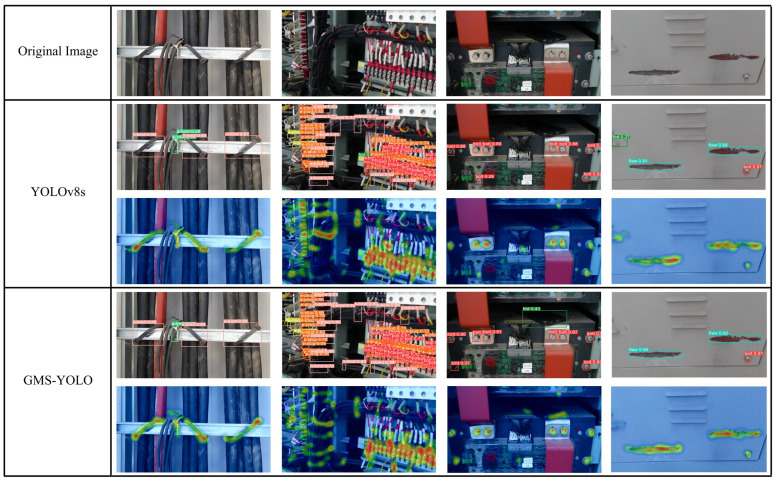
Validation set detection results comparison.

**Table 1 sensors-24-05789-t001:** Detailed dataset data.

Target	Training Set	Validation Set	Total
bolt	14,283	3662	17,945
a-bind	328	84	412
n-bind	8242	2018	10,260
a-plug	352	108	460
e-plug	1296	330	1626
water	1217	302	1519
fod	12,637	2984	15,621
flaw	1191	338	1529
Image Number	4492	1124	5616

**Table 2 sensors-24-05789-t002:** Detailed training configuration.

Laboratory Setting	Configuration Information
CPU	Intel Xeon Gold 6248 R Processor, 24 Core, 48 Thread
GPU	Nvidia GeForce RTX 3090 GPU 24 G
Running System	Windows 10 Professional
Random Access Memory	128 GB (4 × 32 GB)
Read-Only Memory	4.5 TB (512 GB + 4 TB)
Programming Language	Python 3.10
Deep Learning Framework	PyTorch 1.11.0 + CUDA 11.5

**Table 3 sensors-24-05789-t003:** The results of the comparative experiment.

Model	Parameters (M)	Precision (%)	Recall (%)	F1-Score	mAP_50_ (%)	mAP_50:95_ (%)	GFLOPs (G)
YOLOv3-Tiny	12.1	84.7	64.6	0.733	72.5	44.7	19.0
YOLOv5s	9.1	87.5	75.3	0.809	82.1	56.4	24.0
YOLOv6s	16.3	86.3	72.9	0.790	79.5	54.9	44.1
YOLOv7-Tiny	6.0	82.4	73.1	0.775	78.5	47.2	13.2
YOLOv8s	11.1	87.8	77.0	0.820	82.7	57.2	28.5
RT-DETR-l	28.5	79.8	72.4	0.759	77.6	50.3	100.7
RT-DETR-Resnet50	42.0	82.9	73.6	0.780	79.1	52.2	125.8

**Table 4 sensors-24-05789-t004:** Comparisons with other improved algorithms.

Model	Parameters (M)	Precision (%)	Recall (%)	F1-Score	mAP (%)	GFLOPs (G)
MFI-YOLO	9.2	87.6	74.8	0.807	81.6	22.4
CS-YOLO	21.8	92.0	77.0	0.838	84.2	48.4
BL-YOLOv8	7.8	90.4	74.9	0.819	82.5	25.4
Insu-YOLO	4.2	86.9	72.6	0.791	80.6	13.8
GMS-YOLOv5s	5.8	88.6	75.4	0.815	83.3	18.1
GMS-YOLO(ours)	6.9	91.1	78.0	0.840	85.0	20.6

**Table 5 sensors-24-05789-t005:** The results of ablation experiments based on specific categories.

Module			AP (%)							
**GhostHG** **Netv2**	**MSCA**	**SCDH**	**Bolt**	**a-Bind**	**n-Bind**	**a-Plug**	**e-Plug**	**Water**	**Fod**	**Flaw**
✕	✕	✕	77.8	86.7	94.1	72.9	91.4	89.0	90.8	61.8
✔	✕	✕	78.2	88.9	93.7	76.4	90.7	88.7	90.5	62.7
✕	✔	✕	78.0	87.2	94.0	73.6	92.0	89.8	90.8	60.6
✕	✕	✔	78.5	89.0	93.9	75.9	91.6	90.7	91.5	60.8
✔	✔	✕	78.8	88.8	93.8	76.1	92.0	90.4	91.4	61.2
✔	✕	✔	78.4	88.8	93.6	75.7	90.5	89.5	91.1	63.2
✕	✔	✔	78.9	89.1	94.0	75.3	92.8	91.3	91.6	61.9
✔	✔	✔	78.8	91.2	94.0	75.6	92.0	90.9	90.8	66.0

**Table 6 sensors-24-05789-t006:** The results of ablation experiments based on the whole.

Module			Parameters (M)	Precision (%)	Recall (%)	F1-Score	mAP_50_ (%)	mAP_50:95_ (%)	GFLOPs (G)
GhostHGNetv2	MSCA	SCDH							
✕	✕	✕	11.1	87.8	77.0	0.820	82.7	57.2	28.5
✔	✕	✕	8.3	90.1	76.8	0.829	83.8	58.0	23.0
✕	✔	✕	11.4	88.6	78.7	0.834	83.3	58.2	28.8
✕	✕	✔	9.4	90.9	76.4	0.830	84.0	58.7	25.9
✔	✔	✕	8.6	90.0	78.0	0.836	84.2	58.7	23.3
✔	✕	✔	6.6	88.8	76.4	0.821	83.9	58.4	20.4
✕	✔	✔	9.8	89.3	77.4	0.829	84.4	59.0	26.1
✔	✔	✔	6.9	91.1	78.0	0.840	85.0	59.5	20.6

## Data Availability

The original contributions presented in this study are included in the article; further inquiries can be directed to the corresponding author.

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
