# Peer review of "GMS-YOLO: An Algorithm for Multi-Scale Object Detection in Complex Environments in Confined Compartments"

_sensors, 2024, doi:10.3390/s24175789_

Round 1

Reviewer 1 Report

Comments and Suggestions for Authors

Minor revision

The current manuscript elaborates on the detection of multiple objects using an innovative GMS-YOLO network, which builds upon the enhanced YOLOv8 framework, within complex inspection environments. It also provides a thorough comparison of evaluation metrics, complete with detailed descriptions, analyses, and discussions. Although the findings of this study are significant, there are several areas within the document that require improvement to align with the rigorous standards expected for publication. For details, please see attachment.

(1) Dataset Collection - The manuscript could benefit from mentioning instance-augmented image augmentation methods utilized in this study.

(2) Evaluation Indicators - Incorporation of the F1-score into the evaluation metrics would provide a more comprehensive analysis.

(3) Testing Results and Figures - It would be advantageous to include loss curves for bounding box regression and comparative analyses of feature heatmap figures.

(4) Improved Framework - The manuscript discusses the lightweight performance of YOLOv5s and YOLOv7-Tiny in terms of parameters, as shown in Table 3. A comparison between the enhanced versions of GMS-YOLOv5s and GMS-YOLOv7 could be considered.

Author Response

Dear Reviewer:

First and foremost, we would like to express our sincere gratitude for your valuable comments! We have engaged in thorough discussions and meticulous revisions regarding each piece of feedback, and have gained significantly from this process. Below, we will address each comment individually and make corresponding revisions to the manuscript, in the hope of gaining your approval.

Comments 1:

Dataset Collection -The manuscript could benefit from mentioning instance-augmented image augmentation methods utilized in this study.

Response 1:

We have mentioned in the paper.(Page 8-9)

Comments 2:

Evaluation Indicators -Incorporation of the Fl-score into the evaluation metrics would provide a more comprehensive analysis.

Response 2:

We have incorporated the F1-score into the evaluation metrics for all experiments.(Page 10-12, 14)

Comments 3:

Testing Results and Figures - It would be advantageous to include loss curves for bounding box regression and comparative analyses of feature heatmap figures.

Response 3:

We have incorporated comparisons of loss curves and heatmaps in the paper.(Page 14-15)

Comments 4:

Improved Framework -The manuscript discusses the lightweight performance of YOLOv5s and YOLOv7-Tiny in terms of parameters, as shown in Table 3. A comparison between the enhanced versions of GMS-YOLOv5s and GMS-YOLOv7 could be considered.

Response 4:

We have included experiments with GMS-YOLOv5s in comparative tests with other improved methods. However, regarding the YOLOv7 series models, we believe that their performance is somewhat significantly different from YOLOv5 and YOLOv8, making the value of improvement and comparison relatively low. Therefore, we did not conduct corresponding experiments.(Page 11-12)

Comments 5:

The authors may add key information on the figures.

Response 5:

We have added relevant figure descriptions.(Page 10)

Comments 6:

Results Analysis - The AP for the bolt, a-plug, and flaw, as detailed in Table 5, along with the recall rates in Table 6, are notably lower than those for other objects. An in-depth exploration of the reasons behind these results and potential improvements would be beneficial.

Response 6:

We have added relevant analyses in the experimental analysis section.(Page 13)

Comments 7:

The manuscript would benefit from incorporating additional state-of-the-art computer vision articles to enhance its comprehensiveness: Mask YOLOv7-Based Drone Vision System for Automated Cattle Detection and Counting;Artificial Intelligence and Applications. A lightweight improved YOLOv5s model and its deployment for detecting pitaya fruits in daytime and nighttime light-supplement environments; Computers and Electronics in Agriculture. Real-Time Human Detection and Counting System Using Deep Leaming Computer Vision Techniques; Artificial Intelligence and Applications.)

Response 7:

We have incorporated relevant descriptions and references in the introduction section.(Page 2)

Reviewer 2 Report

Comments and Suggestions for Authors

1.Although the experimental results showed that GMS-YOLO was superior to the original model in multiple indicators, it did not fully demonstrate significant differences from existing technologies (in Table 4: only partial improvement compared to the comparison object, improvement of 5% compared to the lightest Insu-YOLO, and improvement of no more than 3% compared to BL-TOLOv8 of the same magnitude, both of which are MDPI paper).

2.There are still some issues with the experimental design. The dataset used in the experiment was not detailed in terms of its composition and preprocessing steps, and the diversity and complexity of image samples captured in confined spaces were not fully demonstrated.

3.The formatting of the paper needs to be more standardized. The content in the text and the insertion position of the legends and charts are too far apart. For example, the last line on page 8 mentions Figure 8, but the figure that connects to the next page is Figure 7, and the Figure 8 is inserted on page 10.  

Comments on the Quality of English Language

Good English expression, able to express clearly, but there are some grammar errors

Author Response

Dear Reviewer:

First and foremost, we would like to express our sincere gratitude for your valuable comments! We have engaged in thorough discussions and meticulous revisions regarding each piece of feedback, and have gained significantly from this process. Below, we will address each comment individually and make corresponding revisions to the manuscript, in the hope of gaining your approval.

Comments 1:

Although the experimental results showed that GMS-YOLO was superior to the original model in multiple indicators, it did not fully demonstrate significant differences from existing technologies (in Table 4: only partial improvement compared to the comparison object, improvement of 5% compared to the lightest Insu-YOLO, and improvement of no more than 3% compared to BL-TOLOv8 of the same magnitude, both of which are MDPI paper).

Response 1:

For Insu-YOLO, although our approach may have some shortcomings in terms of model size, it boasts an absolute advantage in detection performance, which is particularly crucial for inspection tasks where detection accuracy is paramount. As for BL-TOLOv8, our method not only leads in accuracy but also demonstrates significant advantages in both the number of model parameters and computational requirements.

Comments 2:

There are still some issues with the experimental design. The dataset used in the experiment was not detailed in terms of its composition and preprocessing steps, and the diversity and complexity of image samples captured in confined spaces were not fully demonstrated.

Response 2:

We have supplemented the construction process and specific quantities of the dataset in our paper.(Page 8-10) The complexity of images in confined cabins has also been mentioned. The images in the paper illustrate that within limited space, it is necessary to detect multiple categories and quantities of objects, including issues such as object occlusion and insufficient lighting in certain areas. These factors collectively demonstrate the complexity of the detection scenario.

Comments 3:

The formatting of the paper needs to be more standardized. The content in the text and the insertion position of the legends and charts are too far apart. For example, the last line on page 8 mentions Figure 8, but the figure that connects to the next page is Figure 7, and the Figure 8 is inserted on page 10.

Response 3:

We have revised the formatting of the paper, ensuring that any mentioned figures and tables are now located within no more than one page from their reference.